

# Validation of the sea ice surface albedo scheme of the regional climate model HIRHAM–NAOSIM using aircraft measurements during the ACLOUD/PASCAL campaigns

Evelyn Jäkel[1], Johannes Stapf[1], Manfred Wendisch[1], Marcel Nicolaus[2], Wolfgang Dorn[3], and Annette Rinke[3]

[1]Leipzig Institute for Meteorology (LIM), University of Leipzig, Germany
[2]Alfred Wegener Institute Helmholtz Centre for Polar and Marine Research, Bremerhaven, Germany
[3]Alfred Wegener Institute Helmholtz Centre for Polar and Marine Research, Potsdam, Germany

**Correspondence:** Evelyn Jäkel (evi.jaekel@uni-leipzig.de)

**Abstract.** For large scale and long term Arctic climate simulations appropriate parameterization of the surface albedo are required. Therefore, the sea ice surface (SIS) albedo parameterization of the coupled regional climate model HIRHAM–NAOSIM was examined against measurements performed during the joint ACLOUD (Arctic CLoud Observations Using airborne measurements during polar Day) and PASCAL (Physical feedbacks of Arctic boundary layer, Sea ice, Cloud and AerosoL) campaigns which were performed in May/June 2017 north of Svalbard. The SIS albedo parameterization was tested using measured quantities of the prognostic variables surface temperature and snow depth to calculate the surface albedo and the individual fractions of the ice surface subtypes (snow covered ice, bare ice, and melt ponds) derived from digital camera images taken onboard of the Polar 5/6 aircraft. Based on data gained during 12 flights, it was found that the range of parameterized SIS albedo for individual days is smaller than that of the measurements. This was attributed to the biased functional dependence of the SIS albedo parameterization on temperature. Furthermore, a temporal bias was observed with higher values compared to the modeled SIS albedo (0.88 compared to 0.84 for 29 May 2017) in the beginning of the campaign, and an opposite trend towards the end of the campaign (0.67 versus 0.83 for 25 June 2017). Furthermore, the surface type fraction parameterization was tested against the camera image product which revealed an agreement within 1 %. An adjustment of the variables, defining the parameterized SIS albedo, and additionally accounting for the cloud cover could reduce the root mean squared error from 0.14 to 0.04 for cloud free/broken cloud situations and from 0.06 to 0.05 for overcast conditions.

## 1 Introduction

Arctic amplification is significantly driven by the snow/ice albedo feedback. The reduction in snow/ice cover results in a decrease of surface albedo, which enhances the solar heating of the surface due to more absorption of solar radiation at the surface, leading to a further decrease of snow and ice cover (Schneider and Dickinson, 1974; Curry et al., 1995). In particular, in spring , when the solar insolation is rapidly increasing, the Arctic climate system is highly sensitive to changes of sea ice cover (Groisman et al., 1994; Hall, 2004; Déry and Brown, 2007). Pithan and Mauritsen (2014) quantified the strength of various





feedback mechanisms contribution to Arctic amplification using climate simulations from the Coupled Model Intercomparison Project Phase 5 (CMIP5; Taylor et al., 2012) and found that the snow/ice albedo feedback is the second main contributor. Studies as presented by Pithan and Mauritsen (2014) show the need of models to represent the individual processes and feedback mechanisms that contribute to Arctic amplification. In general, these feedbacks are qualitatively captured in current

climate models, but their magnitude and relative contributions are quite uncertain (Qu and Hall, 2014; Fletcher et al., 2015).

In particular, the spread of climate model results with respect to the snow/ice albedo feedback has been discussed (Qu and Hall, 2014; Thackeray and Fletcher, 2016; Thackeray et al., 2018) . Exemplarily, Thackeray et al. (2018) calculated the sensitivity of snow-covered surface albedo ($\alpha$) to surface temperature ($T_\mathrm{surf}$) in terms of $\Delta\alpha/\Delta T_\mathrm{surf}$ based on the CMIP5 model results, and compared the model output with estimates from satellite observations and reanalysis data. They found a

range of -0.67 % K$^{-1}$ > $\Delta\alpha/\Delta T_\mathrm{surf}$ > -1.26 % K$^{-1}$ compared to an observed value of -1.22 % K$^{-1}$. Qu and Hall (2014) derived a range of $\Delta\alpha/\Delta T_\mathrm{surf}$ between -0.46 % K$^{-1}$ to -1.37 % K$^{-1}$ for a similar study based on CMIP5 models. They concluded that the parameterization of snow and ice surface albedo in several models contributes significantly to the bias in the magnitude of the feedback, as a result of the strong sensitivity of simulated snow-covered surface albedo to surface warming. Flanner et al. (2011) and Crook and Forster (2014) have identified an underestimation of the snow/ice albedo feedback in models compared

to observations.

In the Arctic, $\Delta\alpha/\Delta T_\mathrm{surf}$ is primarily related to the reduction of snow/ice cover leading to a decrease of surface albedo, mostly affected by the change of snow properties caused by snow metamorphism processes (Fletcher et al., 2012). Therefore, the representation of the evolution of snow cover and surface albedo, in particular in the melting period, is crucial to obtain reliable estimates from climate models. The scale of snow cover variations is significant on meter-scales, much smaller than

common grid sizes of climate models, therefore, snow/ice cover fractions are parameterized in models, often as function of snow depth or mass. With respect to snow albedo, most climate models assume constant values for fresh and old snow with some assumptions considering the transition between both extremes (linear or polynomial dependency). Either the snow albedo change dependends on time since last snowfall and snow age, or a relation between temperature and snow albedo is applied to account for snow property changes. Pirazzini (2009) and Thackeray et al. (2018) gave a comprehensive overview of the

different assumptions describing snow albedo and cover in climate models.

The CMIP5 model spread in the representation of the sea ice surface (SIS) albedo directly affects the estimates of the cloud radiative forcing (CRF) as shown by Karlsson and Svensson (2013). In particular in summer months, the strong dependence of the CRF from the surface albedo may even lead to different signs in modelled CRF. In fact, during the summer, the differences in SIS albedo contributes more to the model spread in CRF estimates than cloud fraction (Karlsson and Svensson, 2013).

A careful validation of the model output with observations is mandatory. Snow/ice albedo and cover are mainly validated against satellite observations (e.g., Qu and Hall, 2014; Zhou et al., 2014; Fletcher et al., 2015; Verseghy et al., 2017; Thackeray et al., 2018). This is an appropriate method to compare the model output for periods over several years documenting annual and seasonal changes deduced from observations and climate models. However, the variety of representations of the snow/ice albedo and cover in these models represents a main reasons for the spread of the model output. Therefore, a direct validation of

the surface albedo parameterizations can reveal shortcomings of the applied assumptions. To apply these parameterizations, de-



coupled from the model itself, the input parameters (e.g., surface temperature, snow age) have to be provided by measurements. Satellite observations are limited in their spatial and temporal resolution. The latter issue gets important during rapid melting events, which cannot be adequately documented by satellite-based albedo measurements, commonly accumulated over several days as the 16-day MODIS (Moderate Resolution Imaging Spectroradiometer) product (Wang et al., 2014). Furthermore,

optical surface observations by satellites are restricted to cloud free situations.

On a local scale, Curry (2001) and Køltzow (2007) applied various surface albedo parameterization schemes on the one year data set of ground-based observations collected during the Surface Heat Budget of the Arctic Ocean project (SHEBA, Persson et al., 2002). They found that the seasonal cycle of the surface albedo is not-well represented by most of the parameterizations. In particular, the temperature-dependent schemes calculated too low surface albedo values in the transition period between

spring and summer. Pedersen and Winther (2005) investigated the performance of different surface albedo schemes based on measurements performed over 59 years at eight ground sites located in Russia, France, and on Svalbard. On average, the modeled snow surface albedo exhibited smaller values than the measurements. Overall, the local mean root mean squared error (RMSE) between the observed and modeled albedo ranged between 0.09 and 0.15 for the individual models. Pedersen and Winther (2005) emphasized the need for comparisons on larger spatial scales. On model grid scales this can be only achieved

against satellite observations.

In this paper, a comparison is performed on an intermediate spatial scale. Aircraft and ground-based observations taken during the concurrent ACLOUD (Arctic CLoud Observations Using airborne measurements during polar Day) and PASCAL (Physical feedbacks of Arctic boundary layer, Sea ice, Cloud and AerosoL) campaigns (Wendisch et al., 2018) are used to validate the SIS albedo scheme of the coupled regional climate model HIRHAM–NAOSIM (Dorn et al., 2018). Both campaigns

were performed north of Svalbard during the spring–summer transition in 2017. As typical for climate models, the snow/ice albedo is parameterized as a function of surface temperature, whereas snow cover fraction is related to the snow depth. In Section 2, the measured data set (surface albedo, temperature, snow depth, snow/ice fraction) and the parameterization scheme are presented. The validation of the surface albedo and ice/snow cover fraction parameterization was performed for several flights under different sun illumination conditions. A resulting adjustment of the SIS albedo parameterization in HIRHAM–

NAOSIM for sea ice surfaces is given in Section 3.

## 2   Sea ice surface albedo scheme and measurements

### 2.1   Sea ice albedo scheme of HIRHAM–NAOSIM

The most recent version of the coupled regional climate model HIRHAM–NAOSIM consists of the atmosphere component HIRHAM5 and the ocean-sea ice component NAOSIM in its fine-resolution version. A detailed model description is given

by Dorn et al. (2018). Using a regional model with focus on the Arctic allows an improved description of Arctic processes and feedbacks between atmosphere, sea ice, and ocean (Rinke et al., 2013). However, as stated by Køltzow (2007), the model sensitivity to changes in the SIS albedo parameterization would be higher in such coupled climate models, which may lead to larger uncertainties.





The latest version of the SIS albedo scheme applied in HIRHAM–NAOSIM is described by Dorn et al. (2009). Here only the essential equations are presented, which are directly applied on the measured data set. For an inhomogeneous surface, the surface albedo in a model grid cell can be considered as a sum of the individual surface albedo values weighted by the areal fractions ($c$) of the respective surface subtypes:

$$\alpha = c_i \cdot \alpha_i + (1 - c_i) \cdot \alpha_{ow} \quad , \tag{1}$$

with the subscript i indicating the sea ice types, and ow representing open water. Since the surface albedo of sea ice is highly variable compared to open water, which is assumed to be 0.1, the individual albedo of ice types needs to be classified in subtypes: snow-covered ice (subscript s), bare ice (subscript bi) and melt ponds (subscript m) following the parameterization of Køltzow (2007). This results in an overall SIS albedo calculated by:

$$\alpha_i = c_s \cdot \alpha_s + c_m \cdot \alpha_m + (1 - c_s - c_m) \cdot \alpha_{bi} \quad . \tag{2}$$

The albedo of the surface subtypes is variable due to their changing physical properties. In particular in the melting season, the surface reflection properties are changing on a daily basis depending on the surface temperature and snow depth. In the SIS albedo scheme applied in HIRHAM–NAOSIM, Dorn et al. (2009) estimated ranges of possible surface albedo values as shown in Table 1.

**Table 1.** Minimum and maximum values of surface albedo of snow covered ice, bare ice, and melt ponds as used in the SIS albedo scheme.

| Ice Subtype | Albedo minimum ($\alpha_{min}$) | Albedo maximum ($\alpha_{max}$) |
| --- | --- | --- |
| Snow covered ice | 0.77 | 0.84 |
| Bare ice | 0.51 | 0.57 |
| Melt ponds | 0.16 | 0.36 |

The parameterized surface albedo of the subtypes is determined by:

$$\alpha = \alpha_{min} + (\alpha_{max} - \alpha_{min}) \cdot f(T_{surf}) \quad , \tag{3}$$

with $f(T_{surf})$ representing the surface temperature dependent function:

$$f(T_{surf}) = \min(1, \max(0, T_{surf}/T_d)) \quad , \tag{4}$$

where $T_{surf}$ and $T_d$ are given in degrees Celsius. $T_d$ describes a temperature threshold, where the surface albedo change of the subtypes is expected to be significant. In HIRHAM–NAOSIM the value of $T_d$ is set to -0.01 °C for snow covered and bare ice, while for melt ponds $T_d$ is estimated with -2 °C (Dorn et al., 2009). This results in a sharp drop of albedo for bare ice and snow covered ice from $\alpha_{max}$ to $\alpha_{min}$ between $T_{surf}$ = -0.01 and 0.0 °C, whereas for melt ponds a linear decrease of $\alpha_m$ between $T_{surf}$ = -2.0 and 0.0 °C is estimated.





After parameterization of the sea ice subtype surface albedo, their areal fractions according to Eq. (2) need to be retrieved to calculate the final SIS albedo of the model grid cell. The discrimination between bare ice and snow covered sea ice is estimated by the prognostic variable snow thickness ($h_s$) in HIRHAM–NAOSIM. The parameterization of the snow cover fraction as presented by Dorn et al. (2009) reads:

$$c_s = c_{s,max} \cdot \tanh\left(\frac{h_s}{h_{0.75}}\right) \quad , \tag{5}$$

where $c_{s,max}$ is the maximum snow cover fraction of 0.99 and $h_{0.75}$ gives the snow thickness at which 75 % of the sea ice is covered by snow, which is estimated with $h_{0.75} = 0.03$ m. The melt pond fraction is parameterized by:

$$c_m = c_{m,max} \cdot (1 - f(T_{surf})) \quad , \tag{6}$$

as suggested by Køltzow (2007) with $c_{m,max} = 0.22$ giving the maximum melt pond fraction.

The parameterized SIS albedo for a sea ice cover of 100 % is illustrated in Figure 1a. It comprises the temperature dependence of the subtype's surface albedo (Eq. 3) and the melt pond fraction (Eq. 6), as well as the snow depth dependence of the surface fraction of snow covered ice (Eq. 5). The significant gradient of the surface albedo around 0 °C is mainly related to the choice of the small temperature range where the surface albedo of snow covered ice and bare ice migrates from the maximum to the minimum value within $\Delta T_{surf} = 0.01$ K. Outside the temperature range between threshold temperature $T_d$ and 0.0 °C, the parameterized SIS albedo is only dependent on the snow depth, which is directly linked to $c_s$. The change in the surface subtype fractions, exemplarily determined for -0.1 °C, is shown in Figure 1b. The bare ice fraction ($c_{bi} = 1 - c_s - c_m$) is only dominating when snow depth values are lower than 0.02 m. Together with Figure 1a, it becomes obvious that the snow depth dependence of the surface subtype fraction gets solely relevant for $h_s < 0.1$ m.

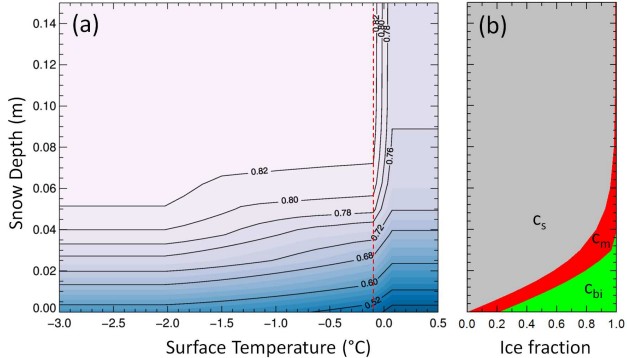

**Figure 1.** (a) Contour plot of the SIS albedo dependent on snow depth and surface temperature as parameterized from the SIS albedo scheme of HIRHAM–NAOSIM for an area with 100 % sea ice cover. The vertical red-dotted line marks a surface temperature of -0.1 °C, for which the surface subtype fractions are plotted in (b).



## 2.2 Ground-based and aircraft observations

To test the SIS albedo scheme of the HIRHAM–NAOSIM model offline from the HIRHAM–NAOSIM output, measured values of the prognostic variables $T_{\text{surf}}$ and $h_{\text{s}}$ are used to calculate the SIS albedo and the individual fractions of the sea ice subtypes. The combined ACLOUD/PASCAL campaigns were conducted in May/June 2017 north of Svalbard as part of the (AC)[3]

project (Wendisch et al., 2017). During ACLOUD, airborne observations of cloud, aerosol particle, and surface properties were collected by the two research aircraft Polar 5 and Polar 6. In connection to the aircraft activities, during PASCAL ground-based measurements on a drifting ice floe station and ship-borne observations with the Research Vessel (RV) Polarstern were conducted in close collocation to the aircraft. Also a data set of the SIS albedo was sampled from buoy observations from the beginning of June to mid of July. An overview of the concurrent campaigns and a synoptic overview is given by Wendisch et al.

(2018) and Knudsen et al. (2018), respectively. Three different synoptic periods were classified during ACLOUD/PASCAL, a cold period (23–29 May 2017), followed by a warm period (30 May to 12 June 2017), and a normal period (13–26 June 2017).

Suitable measurement cases for the validation of the SIS albedo scheme were selected based on the following restrictions: (i) flight altitude lower than $100\,\text{m}$ to minimize atmospheric masking in the surface albedo from the aircraft, (ii) aircraft pitch and roll angle are in a range of $\pm\,4°$, and (iii) no clouds between aircraft and surface. Applying these filters on the aircraft

data, suitable flight sections on 14 flights were identified. The corresponding flight paths of these days from both aircraft are plotted in Figure 2. The percentage sea ice concentration derived for 15 June 2017 from satellite observations by the Advanced Microwave Scanning Radiometer (AMSR) instrument (Spreen et al., 2008) is displayed in the background. As indicator of the variability of the sea ice concentration within the course of the campaign, the $75\,\%$-isolines for 27 May and 26 June are additionally plotted in gray and black color. The most significant decrease of sea ice extension becomes obvious between $8°$

and $12°$ longitude in the Northwest of Svalbard. In May, the sea ice edge was far south in this region, due to northerly winds coupled with a southerly to southwesterly sea ice drift. With the beginning of the warm period at the end of May, the southerly winds led to a north-eastward ice drift. This specific area was mainly observed at the end of May by both aircraft, and mid of June by the Polar 5 aircraft. However, most of the selected flights were conducted over regions with more than $80\,\%$ sea ice coverage.

During a three-weeks period, snow depth was measured along a $3\,\text{km}$ long transect (spatial resolution along track approx. $1\,\text{m}$) over the PASCAL sea ice floe using a Magna Probe (Sturm and Holmgren, 2018). The temporal development of the daily snow depth frequency distribution along the transect is depicted in Figure 3. The mean snow depth decreased rapidly in the course of the campaign as a result of melting in the warm period from $37\pm24\,\text{cm}$ on 5 June to $22\pm18\,\text{cm}$ on 14 June. Mostly relevant for the magnitude of snow albedo is a snow depth below $10\,\text{cm}$, where the albedo reveals an asymptotic increase with increasing snow depth (Grenfell and Perovich, 2004; Perovich, 2007). Considering only the percentage of measurements with

$h_{\text{s}} < 10\,\text{cm}$, an increase from $9\,\%$ on 5 June up to $32\,\%$ on 14 June was observed.

According to Eq. (3), surface temperature data are needed to apply the SIS albedo scheme of HIRHAM–NAOSIM. Onboard of both aircraft, Polar 5 and Polar 6, a nadir pointing infrared sensor (KT19.85, Wendisch et al., 2018) with a field of view of $2°$ was installed to measure the brightness temperature of the surface along the flight track. The instrumental sensitivity



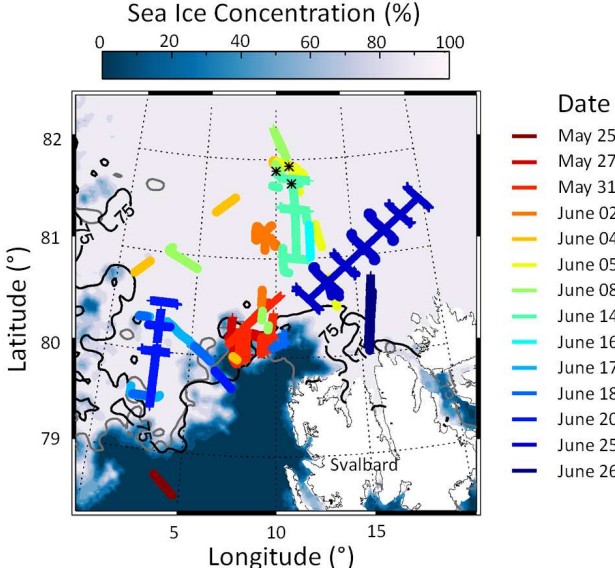

**Figure 2.** Selected flight sections from Polar 5 and Polar 6 during ACLOUD/PASCAL (colored lines). The black stars indicate the area reached by the drifting ice floe between 5 June and 14 June 2017. In the background, the sea ice concentration derived from satellite measurements (AMSR-sensor) for 15 June is shown. Additionally, the 75 %-isolines of the sea ice concentration are plotted for 27 May (in gray) and 26 June (in black).

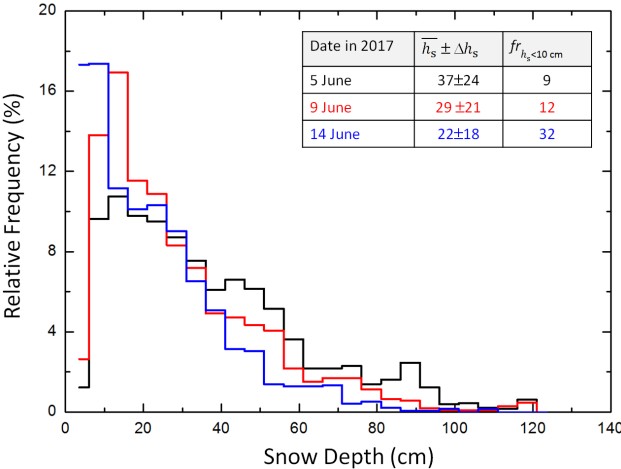

**Figure 3.** Histogram of snow depth in cm measured by the Magna Probe on the ice floe during PASCAL for different days of the period 5–14 June. Additionally, the mean snow depth and the standard deviation, as well as the fraction of measurements with a snow depth below 10 cm ($fr_{h_s < 10cm}$) is given for each day.





covers parts of the atmospheric window between 9.6 $\mu$m and 11.5 $\mu$m wavelength. Within this spectral range, the emissivity of snow and sea ice surfaces varies between 0.965 and 0.995 depending on snow type (Hori et al., 2006). The mean surface temperature and its standard deviation is shown for specific days in Figure 4a. Since Polar 5 was mainly operated above the clouds, less days with suitable cases were found compared to the Polar 6 data set. The increase of surface temperature at the

end of May indicates the change of the synoptic situation (Wendisch et al., 2018). In particular temperatures between -2° and 0°C mid of June represent a crucial range where the albedo of sea ice may change significantly, as considered in Eq. (4). The temporal development of the standard deviation in Figure 4 shows that the variability of surface temperature caused by the contrast between warmer open water and colder sea ice along the individual flight tracks decreases with time.

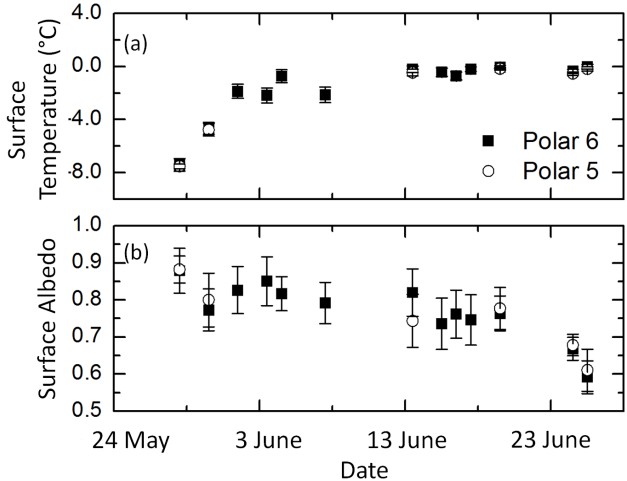

**Figure 4.** (a) Mean surface temperature over sea ice along flight tracks for selected days derived from KT19 measurements onboard of Polar 5 and Polar 6. (b) Mean albedo of sea ice surface. The error bars give the standard deviation.

The effect of the surface temperature increase on the SIS albedo is directly evaluated using concurrent measurements of the

surface albedo by upward and downward looking pyranometers. These pyranometers (Wendisch et al., 2018) were installed on both aircraft to sample the broadband solar irradiance between $0.2 - 3.6$ $\mu$m wavelength. The irradiance data were corrected for aircraft attitude (pitch and roll angle) following the techniques described by Bannehr and Schwiesow (1993). A deconvolution method was applied on the pyranometer measurements to enhance the temporal resolution (20 Hz) of the slow-response sensors (Ehrlich and Wendisch, 2015). The daily mean surface albedo along the selected flight sections as derived from the ratio of

upward and downward irradiances is shown in Figure 4b. A broad range of ice surface albedo values is noticeable between the end of May ($\alpha = 0.86$) and the end of June ($\alpha < 0.6$) as was also shown by Wendisch et al. (2018) (cf. Figure 12 in there). The clearly delayed decrease, caused by the temperature increase, indicates the transformation of sea ice properties.

The fractions of sea ice subtypes were documented by commercial digital cameras on both aircraft (Ehrlich et al., 2012; Wendisch et al., 2018). The cameras were equipped with a 180°-fisheye lens to observe the entire lower hemisphere with a

spatial resolution of 3908 × 2600 pixels. Images were taken every six seconds. Laboratory calibrations of the cameras were





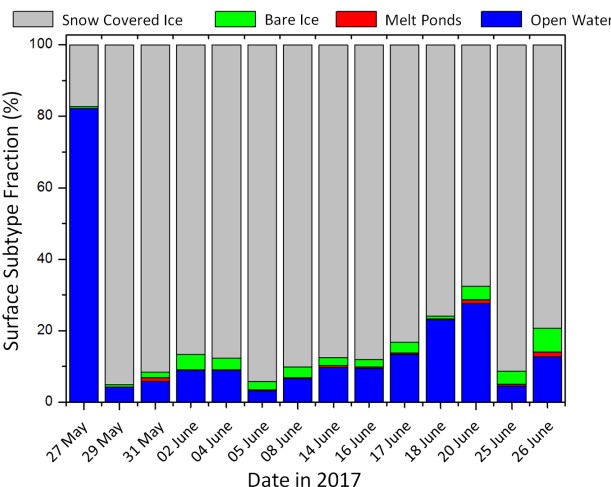

**Figure 5.** Subtype fractions of ice and water surface for selected flight sections during ACLOUD/PASCAL.

performed with respect to their spectral, geometrical, and radiometrical properties. Subsequently, the raw signal for each image pixel was transferred to an absolute radiance value per viewing angle. Techniques to classify the surface types from remote sensing imagery, in particular in Arctic regions, have been introduced by several authors (Perovich et al., 2002; Pedersen et al., 2009; Renner et al., 2013; Divine et al., 2015). In this study, the method of partitioning the image by manually selecting red,

5    green, and blue (RGB) thresholds, based on color intensity histograms, is applied (Perovich et al., 2002). Depending on the illumination conditions, these thresholds were set based on training samples.

The temporal change of the surface subtype fractions derived from Polar 6 measurements along the flight sections, as plotted in Figure 2, is summarized in Figure 5. It shows the daily mean surface subtype fractions of snow covered ice, bare ice, melt ponds, and open water. Note that also flooded sea ice at the sea ice edge might be interpreted as melt ponds by the surface

10   classification method, since these areas exhibit similar spectral features in their reflectectivity as melt ponds. This explains also the occurrence of classified melt pond pixels in the images taken on 31 May.

Since the region of observation is variable (Figure 2), only results of distinct days can be compared with each other. Exemplarily, the mean open water fraction on 31 May was 6 % (Polar 6). The same area was probed on 18 June by the Polar 5 giving a mean open water fraction of 70 %. More northern areas were overflown on 5 June, 14 June, and 16 June. Compared to

15   a 96 % sea ice coverage on 5 June, mid of June 90 % of the area was classified as sea ice. However, the fractions of the sea ice subtypes were almost constant (97 % snow covered ice and 3 % melt ponds and bare ice). Summarized, the surface types were dominated by open water and snow covered ice, whereas melt ponds have only a minor contribution in the probed area during ACLOUD/PASCAL. Thus, the measured quantities are representative for the period describing the beginning of the melting season.





## 3 Validation of the HIRHAM–NAOSIM surface albedo scheme

### 3.1 Procedure

The validation of the SIS albedo scheme of HIRHAM–NAOSIM comprises two components: (a) the surface albedo itself (Sec. 3.2), and (b) the surface subtype fraction parameterization (Sec. 3.3). The procedure is illustrated in Figure 6. The characterization of the surface is based on the digital camera images. For each time step of the images the derived surface albedo from the pyranometers and the surface temperature of the KT19 is selected.

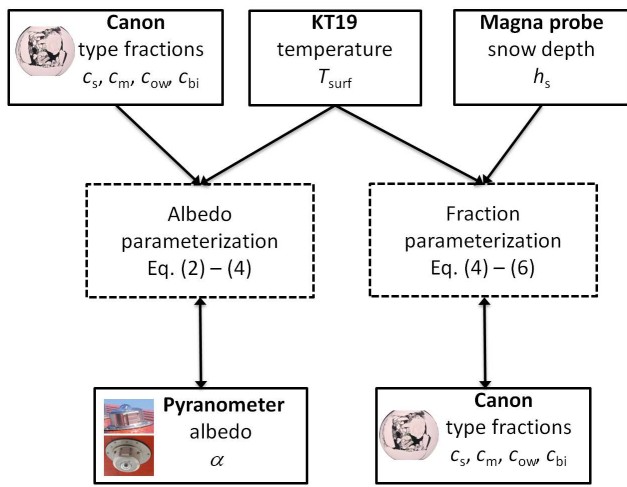

**Figure 6.** Flow chart of validation procedure.

In step (a) the measured surface albedo was compared with the calculated albedo based on the surface subtype fractions deduced from the camera images. $T_{\mathrm{surf}}$ is taken to determine the temperature dependent function $f(T_{\mathrm{surf}})$ according to Eq. (4). Since Eq. (4) will be representative for one scene, only images with 90 % sea ice coverage were selected to omit temperature variations in the field of view of the pyranometer caused by different temperatures of open water and sea ice. Surface subtype fractions derived by the camera were calculated based on two approaches: (i) the summation of area sizes of similar surface types (linear approach), and (ii) similar to (i) but additionally with a weighting by the cosine of the viewing angle. The later approach takes into account that the irradiance is defined as the angular integration of the radiance weighted by the cosine of the viewing angle. This implies that the reflected radiation from side directions has a minor contribution than radiation coming from directly below the aircraft. For a precise validation of Eq. (2) the albedo needs to be measured for all individual surface subtypes close to the surface. Aircraft observations always integrate over a larger area. Therefore, approach (i) might lead to uncertainties due to the cosine weighting. For this reason, the derived surface subtype fractions of both approaches are compared. Mean surface subtype fractions over the different flights were calculated and mean differences not larger than 2 % were found between the two approaches (i) and (ii). The correlation coefficient of the linear regressions between the surface subtype fractions of both approaches were larger than 0.98. Because of this approximate equality the comparison of measured




and parameterized SIS albedo was simplified using only the derived surface subtype fractions, which were based on the linear approach.

Since the digital camera images deliver surface subtype fractions of snow covered ice, melt ponds, bare ice, and open water, the individual subtype fractions have to be adapted for the application of Eq. (2). There, only the fractions of the sea ice subtypes are considered to weight $\alpha_s$, $\alpha_m$, and $\alpha_{bi}$ and to calculate the overall SIS albedo $\alpha_{si}$. Therefore, the surface subtype fractions are rescaled with respect to their total sum.

In step (b) the fraction parameterization of snow covered sea ice and melt ponds (Eqs. (5) and (6)) is validated against estimated surface subtype fractions derived from the digital images. The surface subtype fraction parameterization is based on measured surface temperatures from the KT19 and snow depth data provided from the magna probe. The selection of cases is limited as compared to the SIS albedo scheme validation, caused by the availability of snow depth data in the surrounding of the aircraft flight paths. Only flight sections in the vicinity of the ice floe station were taken into account. Since snow depth is highly variable (Figure 3), the parameterization was applied for the whole distribution of snow depth for the individual days and the measured KT19 temperature. Out of this, the mean parameterized surface subtype fraction for each image time step is compared to the measured fraction of sea ice subtypes. Only scenes with total ice coverage of 95 % were considered. Finally, the SIS albedo parameterization is applied based on the modeled surface subtype fractions from step (b).

## 3.2 Application of the sea ice albedo parameterization

Histograms of the measured and parameterized SIS albedo based on measured surface subtype fractions and surface temperature for individual days during ACLOUD/PASCAL are shown in Figure 7. The days 27 May and 20 June were excluded here, because the number of cases per day was less than 50. The distributions indicated by the thick colored lines, represent the results for a sea ice cover larger than 90 %. Additionally, all cases independent of sea ice coverage are plotted by thin dotted lines. Table 2 summarizes the daily mean measured and parameterized SIS albedo together with their standard deviation of the shown distributions.

The impact of open water within the field of view gets mainly obvious for 18 June and 26 June, where measured and parameterized SIS albedo exhibit also modes for smaller albedo values. For 18 June, the cases showing open water are dominated by subtype fractions $c_{ow} > 50$ %, which results in $\alpha < 0.5$. In contrast, on 26 June, the images indicate the presence of ice floes with $c_{ow}$ lying in the range between 20 % and 30 %, leading to $\alpha > 0.5$.

Two issues become noticeable when comparing the histograms. First, the parameterized SIS albedo shows a narrower distribution than the measured surface albedo. This is especially pronounced for scenes of uniform surface type and surface temperatures which result in $f(T_{surf}) = 0$ according to Eq. (4). In particular, on 29 May temperatures below -4 °C over mostly snow covered ice were measured. This results in a narrow distribution of the SIS albedo, which corresponds to the maximum albedo of snow covered ice $\alpha_{max} = 0.84$ (Table 1). Only with an increase of temperature, when snow melting is considered in the SIS albedo parameterization and when multiple sea ice subtypes are identified in the image, a broader distribution of parameterized albedo is observed (26 June, Figure 7l).





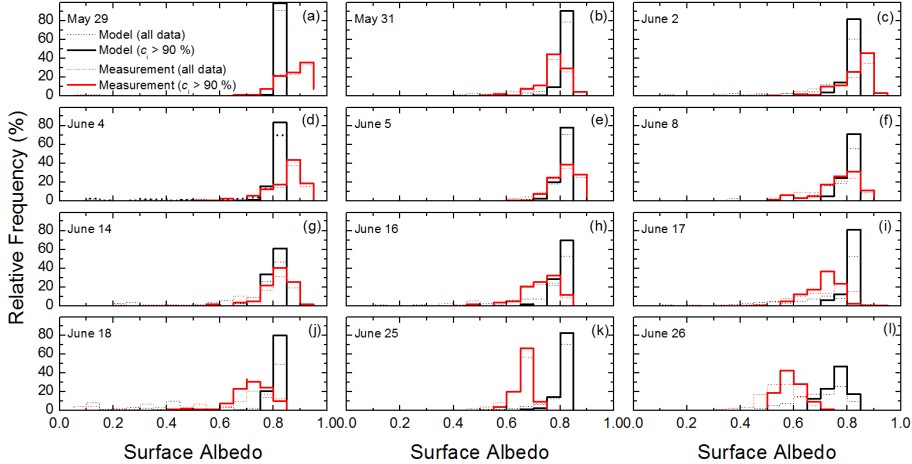

**Figure 7.** (a)-(l) Histograms of measured (red) and parameterized (black) surface albedo for all selected days. Thick lines represent the cases when more than 90 % of the surface is covered by sea ice. Thin dotted lines show all cases matching the selection criteria described in Sec. 2.2.

The second issue obvious from Figure 7 is the temporal change when comparing both SIS albedo distributions. In the beginning of the campaign, when $T_{\mathrm{surf}}$ is lower than -2 °C, the parameterized SIS albedo is systematically lower than the measured one (see also Table 2). In contrast, at the end of June this relation is reversed. An exception of this trend was found for 31 May when the mean measured albedo of sea ice ($\alpha$ = 0.77) was lower than the parameterized albedo ($\alpha$ =

0.83). Sea ice concentration maps from the AMSR sensor show that the ice edge moves further North from 29 May to 31 May, in the area of the flight tracks. This results in more structured snow covered ice for 31 May, as it is also apparent from the digital camera images. Furthermore, different illumination conditions were observed on both days. While on 29 May the SIS albedo was measured under overcast conditions, broken clouds up to clear sky occurred on 31 May. The effect of cloud conditions on the surface albedo was already discussed by several authors before as by Choudhury and Chang (1981) and

Yamanouchi (1983) or more recently by Pirazzini et al. (2015). Clouds tend to increase the proportion of radiation in the visible spectrum. Consequently, the broadband albedo increases under overcast conditions. Effects of the solar zenith angle (SZA) can be excluded here, since for both days the SZA was in the range between 65° and 68°. Thus, the increased roughness together with clear sky conditions might lead to a decrease of the measured SIS albedo. Snow metamorphism leading to larger grain sizes is probably of minor importance, since the surface temperature is below melting temperature ($T_{\mathrm{surf}}$ = -4 °C).

In the HIRHAM–NAOSIM albedo scheme, the parameterized SIS albedo of all surface subtypes only depends on the temperature and the predefined maximum and minimum thresholds of the surface albedo (Eq. (3)). This relation is tested against the measured temperature dependence. For the selected flights, the fraction of melt ponds and bare ice subtypes have never exceeded the 90 % threshold, which was set to assume a homogeneous surface type. Therefore, the evaluation of the relationship between temperature and SIS albedo based on the ACLOUD data set is limited to the snow covered sea ice subtype. All data

were filtered with respect to a fraction of $c_{\mathrm{s}}$ = 100 %. The mean values of surface temperature and SIS albedo for all selected



**Table 2.** Averaged SIS surface albedo and standard deviation (sdv) of each flight retrieved from aircraft observation and parameterization. Additionally, the mean temperature, number of cases, and the illumination conditions are given. Note, that only cases with an ice fraction larger than 90 % are considered here.

| Date in 2017 | Surface Temperature (°) mean±sdv | Measured Albedo mean±sdv | Parameterized Albedo mean±sdv | Number of Cases | Cloud Conditions |
|---|---|---|---|---|---|
| 29 May | -7.4±0.5 | 0.88±0.06 | 0.84±0.01 | 146 | overcast |
| 31 May | -4.7±0.4 | 0.77±0.06 | 0.83±0.02 | 151 | clear / broken cloud |
| 2 June | -1.9±0.5 | 0.83±0.06 | 0.83±0.06 | 157 | overcast |
| 4 June | -2.2±0.6 | 0.85±0.07 | 0.83±0.03 | 189 | overcast |
| 5 June | -0.7±0.5 | 0.82±0.05 | 0.82±0.03 | 272 | overcast |
| 8 June | -2.1±0.5 | 0.79±0.06 | 0.83±0.02 | 85 | clear / broken cloud |
| 14 June | -0.2±0.3 | 0.82±0.06 | 0.82±0.03 | 50 | overcast |
| 16 June | -0.4±0.3 | 0.74±0.07 | 0.82±0.02 | 53 | overcast |
| 17 June | -0.7±0.3 | 0.76±0.06 | 0.82±0.03 | 118 | overcast |
| 18 June | -0.2±0.2 | 0.75±0.07 | 0.82±0.03 | 53 | clear / broken cloud |
| 25 June | -0.4±0.2 | 0.67±0.03 | 0.83±0.02 | 342 | clear |
| 26 June | 0.0±0.2 | 0.59±0.04 | 0.76±0.04 | 64 | clear / broken cloud |

flights is shown in Figure 8a. The data set is separated according to the daily illumination conditions, either overcast (closed symbols) or clear sky / broken cloud (open symbols). Additionally, the corresponding parameterized SIS albedo is marked by the dashed line, revealing the sharp drop of albedo around 0 °C.

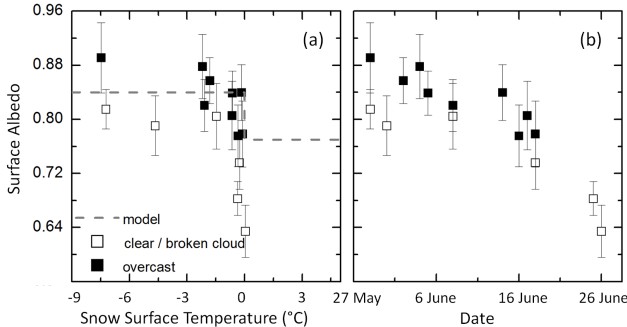

**Figure 8.** Mean albedo of snow covered ice scenes as function of surface temperature (a) and time (b) separated into clear sky / broken clouds and overcast situations. Vertical bars indicate the standard deviation of the averaged surface temperature on each day. The symbols represent the measured albedo and the dashed gray line represents the parameterized albedo.

Figure 8 illustrates that the measured SIS albedo does not follow the surface temperature dependence as assumed by the
5    parameterization. In general, the albedo is decreasing over the early summer with changing temperature. A short term tempera-





ture increase does not necessarily result in a sharp drop of surface albedo as would be predicted by Eq. (3). In fact, the decrease in SIS albedo is mainly caused by a temporal change (Figure 8b) of the surface properties, as grain size and snow thickness, which is not covered by the SIS albedo scheme. Such a temporal decrease of albedo was already observed in Figure 4b with difference that Figure 4b includes also selected scenes of multiple subtypes. Consequently, not only the conversion of surface

subtypes results in the decrease of SIS albedo, but rather the temporal change of snow reflection properties of the snow covered sea ice subtype, which accounts for the largest fraction of all subtypes in the studied cases (see Figure 5).

Besides snow property changes, also the illumination conditions might have an impact on the variation of the surface albedo. Lower SIS albedo values were measured for all cases under cloudless and broken cloud conditions compared to overcast situations with similar surface temperatures ranges. This indicates that the SIS albedo parameterization should also take account

of the dependence on cloud cover fraction.

### 3.3   Application of the surface type classification and full albedo scheme

In this section, the fraction parameterization following Eqs. (5) and (6) is applied including measured snow depth and surface temperature data. While surface temperature data were collocated in time and space with the digital camera observations, the ground-based measured snow depth data were sampled not directly below the aircraft on the selected flight days. Therefore,

the data set was further limited to cases where the aircraft position has not more than 50 km distance from the points of snow depth measurements. In addition, the flight date may differ by one day from the ground-based observation of snow depth. Both restrictions in space and time should ensure the representativeness of local point measurement of the snow depth for the selected area.

Table 3 summarizes the statistics of the measured and parameterized fractions for each day. Here, the median instead of

the daily mean is calculated to exclude the effect of outliers. Furthermore, the distance of the upper and the lower quartiles are given to estimate the range of measured fractions. According to Eq. (5), the snow depth determines the fraction of snow covered ice. The maximum value of $c_s$ of 0.99 is reached for a snow depth larger than 10 cm as defined in Eq. (5). The snow depth distributions in Figure 3 reveal that for all days, apart from 15 June, the majority of snow depth values is larger than 10 cm. Therefore, it is not surprising that the parameterized fractions of snow covered ice is 99 % in all these cases independent

from the shape of the snow depth distribution. The remaining 1 % fraction is either related to melt ponds or bare ice, as defined by the surface temperature. Equation (6) determines the melt pond fractions and indirectly the fraction of the bare ice subtype, because the sum of all three subtypes adds up to one. From the temperatures measured along the selected flight paths, melt ponds were calculated for the days 5 June and 14 June. In summary, the measured and parameterized surface subtype fractions are in very good agreement. However, the variation of surface types is quite low, since the observed region around the ice floe

is covered almost completely by snow. Only in the second half of June, where now snow depth data were measured, the surface type distribution got more complex.

Finally, the calculated fractions as derived above were used as input for Eq. (2) to compute the SIS albedo $\alpha_i$ for all sections connected to the ground-based snow depth measurements on the ice floe. Then, together with measured open water fractions, the final SIS albedo $\alpha$ was derived according to Eq. (1). In Figure 9 the daily averaged albedo values are presented as a box-





**Table 3.** Measured (meas.) and parameterized (par.) subtype fractions of snow covered ice ($c_s$), melt ponds ($c_m$), and bare ice ($c_{bi}$) for selected flight sections on individual days. Q2 stands for the median, Q3-Q1 represent the distance of the upper and lower quartiles. Additionally, the horizontal distance between Polar aircraft and Magna probe is given.

| Date in 2017 Canon data | $c_s$ (meas.) Q2 / Q3-Q1 (%) | $c_s$ (par.) Q2 (%) | $c_m$ (meas.) Q2 / Q3-Q1 (%) | $c_m$ (par.) Q2 (%) | $c_{bi}$ (meas.) Q2 / Q3-Q1 (%) | $c_{bi}$ (par.) Q2 (%) | Date in 2017 snow depth data | Distance (km) |
|---|---|---|---|---|---|---|---|---|
| 4 June | 100.0 / 0.0 | 99.0 | 0.0 / 0.1 | 0.0 | 0.0 / 1.7 | 1.0 | 5 June | 4±5 |
| 5 June | 99.9 / 3.7 | 99.0 | 0.0 / 0.2 | 1.0 | 0.1 / 3.5 | 0.0 | 5 June | 7±8 |
| 8 June | 99.6 / 7.5 | 99.0 | 0.0 / 0.1 | 0.0 | 0.4 / 7.3 | 1.0 | 9 June | 1±2 |
| 14 June | 100. / 0.1 | 99.0 | 0.1 / 0.2 | 1.0 | 0.0 / 0.2 | 0.0 | 14 June | 2±6 |

and-whisker plot. As for the subtype fraction statistics, the median and the quartiles are shown, since the number of cases is quite low (less than ten for 8 June and 14 June) compared to the studied scenes shown in Sec. 3.3 (minimum 50 cases). No variation of the parameterized SIS albedo is found on most of the days, due to the constant relation between the subtype fractions. The variation visible on June 5 and June 8 arises from the variable fraction of open water with a subtype albedo of

$\alpha_{ow} = 0.1$. However, the variability of the measured SIS albedo cannot be fully reproduced by the albedo scheme. The absolute values of measured and parameterized SIS albedo exhibits largest differences for cases which were measured under cloudless / broken cloud conditions (8 June). As the surface temperature dependence function is the main parameter, the resulting modeled SIS albedo for $T_{surf} > 0\,°\mathrm{C}$ corresponds with $\alpha_s = \alpha_{min}$, while on the other days with $T_{surf} < 0\,°\mathrm{C}$, $\alpha_s$ equals $\alpha_{max}$.

### 3.4    Adjustment of the sea ice albedo parameterization

From the 12 flight days during the ACLOUD campaign, differences between modeled and measured SIS albedo ($\alpha_{par}$, $\alpha_{meas}$) were attributed to the impact of illumination conditions (cloudiness) and the choice of minimum and maximum surface albedo values used in Eq. (3). Since uniform subtypes were limited to measurements over snow covered ice, an adjustment of the parameter $\alpha_{max}$, $\alpha_{min}$, and the temperature threshold $T_d$ is only provided for this subtype. All data points with $c_s > 99\,\%$ were separated into two classes depending on the cloud cover, similar to the classification shown in Figure 8. The three parameters

were varied systematically to find the optimum combination with a minimum RMSE as calculated by:

$$\min(\mathrm{RMSE}) = \min\left( \sqrt{\frac{1}{n}\sum_{i=1}^{n}\left(\alpha_{meas,i} - \alpha_{par,i}(\alpha_{max}, \alpha_{min}, T_d)\right)^2} \right) \qquad (7)$$

The final parameters and the corresponding RMSE values for overcast and clear/broken cloud conditions are summarized in Table 4 together with the variation range of $\alpha_{max}$, $\alpha_{min}$, and $T_d$. The minimum and maximum albedo values were tested in a range of 0.5 up to 1.0 in steps of 0.01. The adjusted albedo parameters clearly describe the two cloud conditions with higher

minimum and maximum values (0.80, 0.88) for overcast conditions and lower values (0.66, 0.79) for clear sky and broken cloud situations compared to the suggested numbers given in the original sea ice albedo scheme from Dorn et al. (2009) with





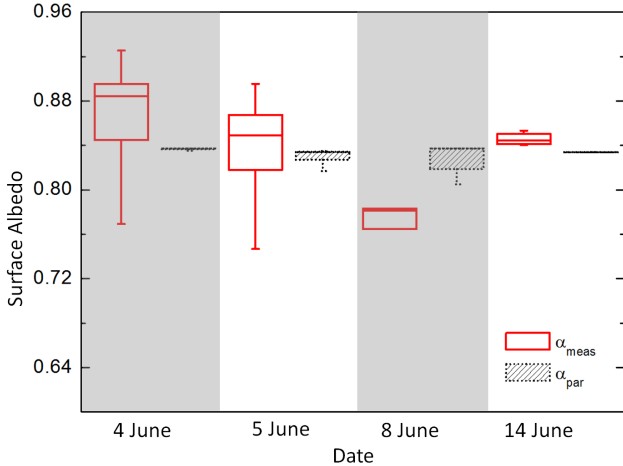

**Figure 9.** Box-and-whisker plot of measured (black solid lines) and parameterized (red dotted lines) surface albedo for selected flight paths in the surrounding of the ice floe where snow depth data were sampled. Minimum and maximum values are displayed as vertical bars. The boxes indicate the 25th, 50th (median), and 75th percentiles of the distribution. The individual days are separted by alternate gray and white areas.

$\alpha_{\mathrm{min}} = 0.77$ and $\alpha_{\mathrm{max}} = 0.84$. Also the threshold temperature was adjusted from -0.01 °C to -2.5 °C (cloudless/broken cloud) and -3.0 °C (overcast). The greatest improvement was found for the parameterization of clear sky surface albedo, where the RMSE values for all cases with $c_{\mathrm{s}} > 99\,\%$ reduced from 0.133 to 0.044, and for all data matching $c_{\mathrm{i}} > 90\,\%$ from 0.141 to 0.041. For overcast situations, the RMSE reduces only slightly from 0.059 to 0.051 for $c_{\mathrm{s}} > 99\,\%$ and from 0.062 to 0.054 for
5   cases with $c_{\mathrm{i}} > 90\,\%$.

**Table 4.** Variation range of minimum and maximum albedo values for snow covered ice and threshold temperature for the adjustment of the sea ice albedo parameterization following Eq. (3) and (4). The albedo and temperature are modified in steps of 0.01 and 0.1°C, respectively. Final fitting values of $\alpha_{\mathrm{min}}, \alpha_{\mathrm{max}}$, and $T_{\mathrm{d}}$ are given for clear/broken cloud and overcast conditions together with the new and old (in brackets) RMSE values.

|  | Clear/broken cloud | Overcast |
|---|---|---|
| Range $\alpha_{\mathrm{min}}$ | 0.50 to 1.00 | 0.50 to 1.00 |
| Range $\alpha_{\mathrm{max}}$ | 0.50 to 1.00 | 0.50 to 1.00 |
| Range $T_{\mathrm{d}}$ (°C) | -5.0 to -0.1 | -5.0 to -0.1 |
| New $\alpha_{\mathrm{min}}$ | 0.66 | 0.80 |
| New $\alpha_{\mathrm{max}}$ | 0.79 | 0.88 |
| New $T_{\mathrm{d}}$ (°C) | -2.5 | -3.0 |
| RMSE ($c_{\mathrm{s}} > 99\,\%$) | 0.044 (0.133) | 0.051 (0.059) |
| RMSE ($c_{\mathrm{i}} > 90\,\%$) | 0.041 (0.141) | 0.054 (0.062) |





## 4   Summary and Conclusion

The parameterizations of sea ice albedo and sea ice subtype fraction as used in the SIS albedo scheme of the coupled regional climate model HIRHAM–NAOSIM were tested with airborne surface albedo, sea ice fraction, and surface temperature measurements taken during the ACLOUD/PASCAL campaign performed north of Svalbard in May/June 2017. The SIS albedo

parameterization requires information on surface temperature and sea ice subtype (snow covered ice, bare ice, and melt ponds) cover fractions. In HIRHAM–NAOSIM, these subtype fractions are calculated from the prognostic variables of surface temperature and snow depth. In this paper, we use corresponding measurements of these parameters, calculate the respective sea ice fractions and surface albedo using the scheme of HIRHAM–NAOSIM, and compare with concurrent measurements.

In a first step, both parameterizations (sea ice albedo and sea ice fraction) were considered separately to compare them with

measurements of surface albedo and surface subtype fraction. In step two, both parameterizations were combined. Based on the measured surface temperature and surface subtype fractions, the SIS albedo was calculated for low-level flight sections which were selected from 12 flights over Arctic sea ice under different illumination conditions.

It was found that (i) the daily histograms of the modeled surface albedo exhibit a narrower distribution than the aircraft measurements, in particular for surface temperatures that are outside of the transition range between dry and melting snow/ice.

Furthermore, (ii) a temporal shift of the deviation between both products was observed with lower modeled SIS albedo compared to measurements in the beginning of the campaign (0.84 vs. 0.88) and higher values derived at the end (0.76 vs. 0.59). Finally, (iii) a dependence of the illumination conditions was observed from the measurements, which are confirmed by former publications (e.g., Choudhury and Chang, 1981; Pirazzini et al., 2015) reporting also lower SIS albedo values for clear sky conditions than in overcast situations.

The subytpe fraction parameterization was applied to ground-based measurements of the snow depth taken on an ice floe for cases when the aircraft position was within a 50 km radius. The modeled surface subtype fractions agreed within 1 % with the measurements, which is also related to the fact that the variability of surface types in the observed area is low with mostly snow covered surfaces. As a result, similar findings were derived when using modeled fractions for the SIS albedo parameterization compared to the decoupled case.

Finally, the SIS albedo parameterization was adjusted for the ACLOUD/PASCAL conditions by defining new values of maximum and minimum surface albedos and threshold temperatures for snow covered sea ice under cloud-free/broken clouds and overcast situations, resulting in deviations to the standard parameters $\Delta\alpha_{\min}$ = +0.03 / -0.11 (overcast/cloud-less), $\Delta\alpha_{\max}$ = +0.04 / -0.05, and a new temperature threshold value of $T_{\mathrm{d}}$ = -3.0°C / -2.5°C compared to $T_{\mathrm{d}}$ = -0.01°C as used in the standard scheme. These adjustments reduced the RMSE from 0.14 to 0.04 for cloud-less/broken clouds and from 0.06 to 0.05

for overcast conditions. The implementation of the adjusted SIS albedo parameterization into HIRHAM-NAOSIM is underway. In the near future, we plan to perform ensemble sensitivity experiments and statistically evaluate the model's skill, including the involved feedback mechanisms.

These results indicate that the correlation between surface temperature and snow surface albedo, and the choice of the minimum and maximum albedo values can serve only as rough estimate of the real albedo. As already shown by Pirazzini



(2009), the daily mean surface albedo dependence on surface temperature exhibits a significant variation and is affected by the surface albedo of underlying sea ice when snow melting has started.

Two main conclusions can be drawn from this study. First considering the dependence on cloud cover fraction in the SIS albedo parameterization, and a second one adjusting the maximum and minimum values of the surface albedo, in particular

for low snow depth. For a snow depth lower than 10 cm, snow albedo is significantly decreasing with decreasing snow depth (Perovich, 2007), which could be considered in an adapted parameterization as proposed by Ahmad and Haider (2015). Both parameters, cloud fraction and snow depth are variables delivered by HIRHAM–NAOSIM.

Furthermore, a larger data set will be necessary to check if the adjustments based on the ACLOUD/PASCAL conditions are valid for other conditions and locations. The upcoming Multidisciplinary drifting Observatory for Studies of the Arctic

Climate (MOSAiC) will provide such an opportunity starting in autumn 2019, when the research vessel *Polarstern* will be drifting with the sea ice for one year supported by two major aircraft campaigns, where the instrumental setup from ACLOUD will be extended by a snow depth radar and a infrared imager, providing an unique data set to validate albedo schemes which are based on surface temperature and snow depth information.

*Competing interests.* The authors declare that they have no conflict of interest.

*Acknowledgements.* We gratefully acknowledge the funding by the Deutsche Forschungsgemeinschaft (DFG, German Research Foundation) – Projektnummer 268020496 – TRR 172, within the Transregional Collaborative Research Center "ArctiC Amplification: Climate Relevant Atmospheric and SurfaCe Processes, and Feedback Mechanisms (AC)[3]. The authors are grateful to AWI for providing and operating the two aircraft during the ACLOUD campaign. We thank the crews of Polar 5 and Polar 6, the technicians of the aircraft for excellent technical and logistical support. The generous funding of the flight hours for ACLOUD by AWI is greatly appreciated. We thank the captain T. Wunderlich

and the cruise leaders A. Macke and H. Flores of the Polarstern expedition PS106 (grant: AWI_PS106_00) for their support in order to enable the ground truthing data from the ice.



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
