# Peer review of "Validation of the sea ice surface albedo scheme of the regional climate model HIRHAM–NAOSIM using aircraft measurements during the ACLOUD/PASCAL campaigns"

_The Cryosphere, 2018_

## Referee Comment (RC1) · Anonymous Referee #1 · 15 Mar 2019

**Jakel et al.: Validation of the sea ice surface albedo scheme of the regional climate model HIRHAM-NAOSIM using aircraft measurements during the ACLOUD/PASCAL campaigns**

The authors use data from the ACLOUD/PASCAL field campaigns in May/June to force a surface albedo parameterization from HIRHAM/NAOSIM and compare to observations. The major findings are that early in the season the parameterized albedos are too low but later they are too high; the parameterized distribution of albedo is too narrow; and the cloud cover affects the observed albedos. I have a few major concerns related to the focus on cloud cover impacts, snow heterogeneity, and surface type.

**Major concerns:**

- Cloud focus: There needs to be more discussion of why cloud cover should impact albedo. Is it due to more scattered light? Additionally, there should be more discussion of how the modifications to albedo for cloud cover would work in a coupled model. Typically, just surface fluxes (SW, LW) are passed to an ice model from the atmosphere model. Could SW or LW be used instead of cloud cover because modeled cloud fraction is notoriously poor.
- Snow on the surface: Nearly all the observations compared are over snow covered ice, but there is little discussion of snow heterogeneity and how this might impact the results. At Pg.8/Ln.2 you mention "snow type" and also later and at Pg.12/Ln.6 you mention "more structured snow" and "increased roughness", all of which allude to the heterogeneity in the snow cover. In fact, Fig.4 (and Pg.8/Ln.15-17) shows that there doesn't seem to be good correlation between changes in temperature, which is relatively constant between early and late June, and albedo, which drops off during this period. Since during this time the ice remains snow covered, this suggests to me that perhaps changes in the snow rather than temperature should be the impetus for the albedo change. Finally, on Pg.14/Ln.2 you mention grain size or snow thickness as being important for temporal evolution on albedo. Why don't you focus on better understanding these snow effects on albedo rather than clouds? Could these snow differences be important for the larger variability in observations than the parameterization? It's known that the snow type is important, it seems worth more mention in this manuscript.
- Other surface types: I am concerned that all the comparisons are done over ice with nearly 100% snow cover. You are clear that the results of the work are valid for covered ice, but later in the season does that mean these results are unimportant? Do clouds have any impact when there are more melt ponds, and is it worth the effort to include cloud cover?

**Minor concerns.** There are just a number of small clarifications or suggestions for figures.

- Fig.1a – Add a colorbar.
- Fig.4 – is there a better way to show this? I can hardly see the whiskers or differentiate between polar 5 and 6 flights. The other thing to point out is that in

panel b there is a large range of observed albedos on each day. This is worth pointing out. Even in early May there are albedos of 0.7 within one standard deviation.

- Fig.6 – can you clarify on the figure which part is a and b of the components for assessment?
- Fig.7 – the dashed lines are very hard to see.
- Fig. 9 – Do red or black correspond to measurements? The caption and legend conflict.
- Table 2 – it looks like for a number of days the parameterized and observed albedos are similar. Is it worth mentioning this?
- Throughout – Does SIS just mean "sea ice surface" or it is the name of the model albedo parameterization.
- Equation 6 – Why is the maximum fraction for melt ponds (22%) so low? Is there justification?
- Pg. 5 Line 17-18: It looks like for hs > 0.1 then the fraction is solely snow-covered ice while for smaller snow depth melt ponds or bare ice become more relevant. I didn't follow the text here.
- Pg.6 Line 20: The winds cause the southwesterly ice drift but your wording is confusing: "due to northerly winds coupled with a southerly to southwesterly sea ice drift."
- Pg.6 Line 30-31: What are the increase of 9% and 32% compared against.
- Pg.9 Line 14: No bars in Fig.5 have 70% open water.
- Pg.10 Line 15: do you mean "coming from directly below the aircraft for (i) only." … "Therefore approach (ii) might lead…".
- Pg.11 Line 19: Why did you use 50 as a threshold?
- Pg.12 Line 14: what is snow grain size differences less important relative to?
- Pg.16 Line 1: the values of albedo given for min and max don't match those in Table 4. I'm confused. Also the precision for RMSE values is probably too great.

---

## Referee Comment (RC2) · Anonymous Referee #2 · 21 Mar 2019

This manuscript describes a field effort (ACLOUD / PASCAL) to validate an established parameterization (HIRHAM–NAOSIM) for the surface albedo of a sea ice cover. The observations mostly consist of snow covered ice, although some bare ice and some ponded ice surfaces are included.

Major concerns: The overwhelmingly dominant surface type for this study is snow-covered ice. Should that be reflected in the title?

The conclusions of this manuscript, as stated p. 18, line 3 provide only limited scientific

insight. The details of the model parameterization and the data set being used to validate it are nicely described, but there is not a lot of fresh scientific insight that results.

Minor points:

Abstract line 5: "The SIS albedo parameterization was tested using measured quantities of the prognostic variables surface temperature and snow depth to calculate the surface albedo and the individual fractions of the ice surface subtypes (snow covered ice, bare ice, and melt ponds) derived from digital camera images taken onboard of the Polar 5/6 aircraft." It would be helpful to include the albedo measurement in this list (broadband? Spectral?).

abstract line 10: "...a temporal bias was observed..." Is this necessarily a temporal bias? It's probably more likely a surface type bias. I doubt the bias depends explicitly on time, but it more likely depends on surface type.

p. 2, line 2: "...the second main contributor." compared to what process?

p. 2, line 6: "the spread of climate model results with respect to the snow/ice albedo feedback has been discussed" can this be made more specific? I think I understand this sentence is trying to convey that the sensitivity of climate model results to parameters directly related to snow/ice albedo feedback are discussed, but this is not clear.

Table 1: Where do the min and max values come from? 0.51 - 0.57 seems like a range that I would expect to be biased low.

p. 10, line 14: "This implies that the reflected radiation from side directions has a minor contribution than radiation coming from directly below the aircraft." Please rewrite for clarity– ". . . has a minor contribution relative to radiation..."?

p.12 line 6: delete "it", also please explain what is meant by "structured snow covered ice".

p. 14, line 2 -3: If I understand correctly, this albedo parameterization does not account for varying grain size and snow depth? That seems like it is important to mention.

p. 14, line 7: "...also the illumination conditions might have an impact on the variation of the surface albedo. Lower SIS albedo values were measured for all cases under cloudless and broken cloud conditions compared to overcast situations with similar surface temperatures ranges." That is expected and it would be helpful to acknowledge that here.

---

## Author Comment (AC1) · 26 Apr 2019

**Reply to Reviewer #1:**

**We thank the reviewer for the time and efforts she/he spent reading our manuscript and providing valuable suggestions and advices. Please find below a discussion of the reviewer's comments (italic). Changes/additions made to the text are underlined and given in quotes.**

*Major concerns:*

*Cloud focus: There needs to be more discussion of why cloud cover should impact albedo. Is it due to more scattered light?*

We added some more explanation:

"Clouds affect the spectral behavior of the incident solar radiation and the directional dependence. In cloudy conditions the incoming radiation field is dominated by the diffuse component, whereas the transmission of radiation through the clouds is wavelength-dependent. Since the solar radiation is mainly absorbed by cloud particles in the near-infrared spectral range, a larger fraction of visible to global radiation is incident on the surface compared to clear sky conditions. Furthermore, the enhanced multiple-scattering between clouds and snow surface additionally contributes to the spectral shift of the incident radiation. Consequently, the broadband albedo increases under cloudy conditions.

Effects of the solar zenith angle (SZA) on the observed differences in Figure 7a,b can be excluded here, since for both days the SZA was in the range between 65° and 68°."

*Additionally, there should be more discussion of how the modifications to albedo for cloud cover would work in a coupled model. Typically, just surface fluxes (SW, LW) are passed to an ice model from the atmosphere model. Could SW or LW be used instead of cloud cover because modeled cloud fraction is notoriously poor.*

We derived adjusted albedo parameters, which clearly reflect the impact of the cloud situation with higher minimum and maximum values (0.80, 0.88) for overcast conditions and lower values (0.66, 0.79) for clear sky and broken cloud situations (Page 16, second paragraph; Page 17, last paragraph). We do argue that it is a solid approach to implement those changes into the coupled model HIRHAM-NAOSIM in a next step. With this we will follow the common approach where albedo modifications are implemented in coupled model (e.g., Rae et al., 2015, doi:10.5194/gmd-8-2221-2015; Koenigk et al., 2011, 10.1007/s00382-011-1132-z). It is the aim to improve the physical description of the albedo, and it is known that the cloud effect needs to be taken into account (see response above). Thus, the next step of implementing this effect is logically. However, we do agree that clouds are generally poorly simulated in the Arctic, but the radiative fluxes are poor too (e.g., English, 2015, doi:10.1175/JCLI-D-14-00801.1). Of course, both are related to each other, and the surface albedo plays an important role (e.g., Karlsson and Svensson, 2013, doi:10.1002/grl.50768). The step afterwards is therefore to further improve the cloud cover simulation. We have already shown that we could improve this by a more efficient Bergeron-Findeisen process and a more generalized subgrid-scale variability of total water content (Klaus et al., 2016, doi:10.1002/2015GL067530). Still, the simulations are far from being perfect and therefore, this is still an ongoing topic for us.

According to your comment, we added a short paragraph in section 4 (Summary and conclusion). It reads (page 18, line 3):

"Although we could improve the cloud cover simulations in HIRHAM5 (Klaus et al., 2016), the simultaneous evaluation of SIS albedo and cloud-radiation (e.g. following Karlsson and Svensson, 2013) in the coupled model HIRHAM-NAOSIM is on our agenda."

*Snow on the surface: Nearly all the observations compared are over snow covered ice, but there is little discussion of snow heterogeneity and how this might impact the results. At Pg.8/Ln.2 you mention "snow type" and also later and at Pg.12/Ln.6 you mention "more structured snow" and "increased roughness", all of which allude to the heterogeneity in the snow cover. In fact, Fig.4 (and Pg.8/Ln.15-17) shows that there doesn't seem to be good correlation between changes in temperature, which is relatively constant between early and late June, and albedo, which drops off during this period. Since during this time the ice remains snow covered, this suggests to me that perhaps changes in the snow rather than temperature should be the impetus for the albedo change. Finally, on Pg.14/Ln.2 you mention grain size or snow thickness as being important for temporal evolution on albedo. Why don't you focus on better understanding these snow effects on albedo rather than clouds? Could these snow differences be important for the larger variability in observations than the parameterization? It's known that the snow type is important, it seems worth more mention in this manuscript.*

We totally agree that the relation of snow property changes controlling the snow albedo is very important. In fact, those snow processes (changes in grain size and density, metamorphism, compacting and ageing, multiple layering, etc.) are commonly covered (by different complexity) in land surface models (e.g., Wang et al., 2016, doi:10.5194/tc-10-1721-2016 and references therein) and ice-ocean models (e.g., Lecomte et al., 2015, doi:10.1016/j.ocemod.2014.11.005; Liston et al., 2018, doi:10.1002/2017JC013706). But, these complex snow processes are still generally only basically covered in sea ice models as part of global coupled climate models (e.g., Hunke et al., 2010, doi:10.3189/002214311796406095), and are currently incorporated only in few global coupled climate models (e.g., Blazey et al., 2013, doi:10.5194/tc-7-1887-2013). New models are currently configured (e.g., Petty et al., 2018, 10.5194/gmd-11-4577-2018). However, also, site-level snow measurements are quite limited over Arctic sea ice and the derivation of reliable snow and ice thickness products from satellite data is still a research in progress.

Actually, we follow both these model and observational developments of these aspects. Accordingly, we will analyse the measured data set which will be gained during the one-year Arctic MOSAiC campaign in 2020.

The albedo variation shown Fig.4 and discussed on Pg.8/Ln.15-17 includes all data along the flight track and consequently comprises also other surface types than snow covered ice (e.g., dark open water). Figure 8 illustrated the temperature and cloud dependence for snow covered ice only. It shows clearly that the snow type variation (change of roughness, grain size, …) is in the same order than the illumination effects. Thus, there is definitely a need to improve the parameterization in this regard too. According to your comment, we added a short paragraph in section 4 (Summary and conclusion). It reads (page 18, line 13):

"Furthermore, our results indicate that the snow type variation (e.g., change of roughness, grain size, density) is of the same order of importance for albedo variations than the illumination (cloud cover) effect. This supports the need to put efforts to improve the snow process parameterizations in coupled models as discussed by Hunke et al. (2010)."

*Other surface types: I am concerned that all the comparisons are done over ice with nearly 100% snow cover. You are clear that the results of the work are valid for covered ice, but later in the season does that mean these results are unimportant? Do clouds have any impact when there are more melt ponds, and is it worth the effort to include cloud cover?*

The reviewer raises an important question here. The adjustments proposed in this work provide improvements for the observed period. It has to be tested how the new parameterization performs for other periods. Also this will be answered based on the MOSAiC observations. We are aware that the importance of melt ponds will increase in the summer season and the adjustments made for the parameters controlling the snow covered ice albedo are of minor importance then, but we expect similar cloud effects on the variation of the melt pond albedo than observed during ACLOUD/PASCAL for snow,

because the physical reason (spectral shift of incident radiation) is also valid for melt ponds which are characterized by a pronounced spectral signature of the albedo between the visible and near-infrared spectral range.

According to your comment, we added a short paragraph in section 4 (Summary and conclusion). It reads (page 18, line 14):

"The presented results are valid for nearly 100% snow covered sea ice. In the later summer season, melt ponds become an important feature. Still, it is expected that the effect of cloud cover on the variation of the melt pond albedo plays a role due to the spectral shift of incident radiation."

*Minor concerns.*

*There are just a number of small clarifications or suggestions for figures.*

*Fig.1a – Add a colorbar.*
Added as suggested:

[Figure]

**Figure 1.** (a) Contour plot of the SIS albedo dependent on snow depth and surface temperature as parameterized from the SIS albedo scheme of HIRHAM–NAOSIM for an area with 100 % sea ice cover. The vertical red-dotted line marks a surface temperature of -0.1 °C, for which the surface subtype fractions are plotted in (b).

*Fig.4 – is there a better way to show this? I can hardly see the whiskers or differentiate between polar 5 and 6 flights. The other thing to point out is that in panel b there is a large range of observed albedos on each day. This is worth pointing out. Even in early May there are albedos of 0.7 within one standard deviation.*
We adjusted the figure (decrease symbol size, colors instead of black/white) for a better separation between individual data points:

[Figure]

**Figure 4.** (a) Mean surface temperature over sea ice along flight tracks for selected days derived from KT19 measurements onboard of Polar 5 and Polar 6. (b) Mean albedo of sea ice surface. The error bars give the standard deviation.

Furthermore we comment the broad standard deviation as follows:

"As indicated by the range of the standard deviation, the spatial variability of the SIS albedo may have the same order of magnitude than the temporal variation."

*Fig.6 – can you clarify on the figure which part is a and b of the components for assessment?*
We adapted the figure by using background colors for separation of the two components:

[Figure]

**Figure 6.** Flow chart of validation procedure of (a) surface albedo (blue background), and (b) surface type fraction (red background) parameterization.

*Fig.7 – the dashed lines are very hard to see.*
We changed the line style representing all data to thin solid lines and adjusted the annotations accordingly:

[Figure]

**Figure 7.** (a)-(l) Histograms of measured (red) and parameterized (black) surface albedo for all selected days. Thick lines represent the cases when more than 90 % of the surface is covered by sea ice. Thinner lines show all cases matching the selection criteria described in Sec. 2.2.

*Fig. 9 – Do red or black correspond to measurements? The caption and legend conflict.*
Thanks for this advice. We fixed the issue by adjusting the figure caption:
"Box-and-whisker plot of measured (red solid lines) and parameterized (black dotted lines) surface albedo for selected flight paths in the surrounding of the ice floe where snow depth data were sampled."

*Table 2 – it looks like for a number of days the parameterized and observed albedos are similar. Is it worth mentioning this?*
We added a short statement:
"In contrast, at the end of June this relation is reversed, while in the transition period the mean parameterized SIS albedo agrees well with the measurements, particularly for overcast cloud conditions."

*Throughout – Does SIS just mean "sea ice surface" or it is the name of the model albedo parameterization.*
The abbreviation SIS was introduced in Section 1 as follows:
"The CMIP5 model spread in the representation of the sea ice surface (SIS) albedo directly affects the estimates of the cloud radiative forcing (CRF) as shown by Karlsson and Svensson (2013)."

*Equation 6 – Why is the maximum fraction for melt ponds (22%) so low? Is there justification?*
According to Køltzow (2007) the threshold temperature for the onset of melt pond development (derived from SHEBA measurements) was set to -2°C. The limitation of the amount of melt pond fraction to 0.22 prevents a complete conversion from snow to melt ponds when temperature is reaching 0°C like observed during ACLOUD. However, the given number of 0.22 refers to SHEBA measurements but is not further discussed in the publication by Køltzow (2007). We are aware that the melt pond fraction may span a larger range than assumed in the parameterization, in particular for July and August (e.g., Istomina et al., 2015).

*Pg. 5 Line 17-18: It looks like for hs > 0.1 then the fraction is solely snow-covered ice while for smaller snow depth melt ponds or bare ice become more relevant. I didn't follow the text here.*
The reviewer is right. For $h\_s > 0.1$ m no other ice types are modelled when temperature is lower than 0°C. Fig. 1b illustrates exemplarily the subtype fraction for T=-0.1°C. The fraction of bare ice is only dominating (50% of total ice fraction) when snow depth is lower than 0.01 m for this specific temperature. We changed the number in the text:
"The bare ice fraction (c_bi=1-c_s-c_m) is only dominating when snow depth values are lower than 0.01 m for this specific case."

*Pg.6 Line 20: The winds cause the southwesterly ice drift but your wording is confusing: "due to northerly winds coupled with a southerly to southwesterly sea ice drift."*
We changed the wording:
*"*In May, the sea ice edge was far south in this region, due to northerly winds a southerly to southwesterly sea ice drift was observed. With the beginning of the warm period at the end of May, the southerly winds led to a north-eastward ice drift (Wendisch et al., 2018)."

*Pg.6 Line 30-31: What are the increase of 9% and 32% compared against.*
We changed the wording:
*"Considering only the percentage of measurements with hs < 10 cm, revealed an increase of this fraction on the overall snow depth observations from 9 % on 5 June to 32 % on 14 June."*
Furthermore, we added the units in Fig. 3 (insert table):

[Figure]

**Figure 3.** Histogram of snow depth in cm measured by the Magna Probe on the ice floe during PASCAL for different days of the period 5–14 June. Additionally, the mean snow depth and the standard deviation, as well as the fraction of measurements with a snow depth below 10 cm ($fr_{h_s<10cm}$) is given for each day.

*Pg.9 Line 14: No bars in Fig.5 have 70% open water.*
Figure 5 only shows the fractions of Polar 6 measurements. The 70 % of open water fraction refers to Polar 5 observations. To make it clearer, we adjusted the figure caption and modified the sentence slightly:

**Figure 5.** Subtype fractions of ice and water surface for selected flight sections of Polar 6 during ACLOUD/PASCAL.
"The same area was probed on 18 June by the Polar 5 giving a mean open water fraction of 70 % (not shown)."

*Pg.10 Line 15: do you mean "coming from directly below the aircraft for (i) only." … "Therefore approach (ii) might lead…".*
The parameterized albedo in one model grid is the mean of the albedo of all subtypes weighted by their fraction of subtype occurrence (Eq. 2). In contrast, the measured albedo along the flight track is additionally depending on the cosine weighting because of the definition of the quantity irradiance. Therefore, we compared the non-weighting (i) and the cosine weighting approach (ii). We revised the sentences:
"This implies that the reflected radiation from side directions has a minor contribution relative to the radiation coming from nadir direction. […]Therefore, approach (i) might lead to uncertainties due to the neglect of cosine weighting."

*Pg.11 Line 19: Why did you use 50 as a threshold?*
The minimum sample (n) size can be approximated by:
$$n \geq \frac{z^2 \sigma^2}{e^2}$$
with z: confidence (95% → z ≅ 2), $\sigma^2$: variance, and *e:* assumed precision of the mean albedo. Taking the measured variance ($0.07^2$) and the desired albedo uncertainty (0.02) into account, *n* needs to be larger than 49.

*Pg.12 Line 14: what is snow grain size differences less important relative to?*
The comparison is related to the roughness and illumination effect, mentioned the sentence before. We connected both sentences now:
"Thus, the likely dominating effect of the clear sky conditions together with the increased roughness lead to a decrease of the measured SIS albedo, whereas the snow metamorphism causing larger grain sizes is

probably of minor importance, since the surface temperature is below melting temperature (Tsurf = -4°C).”

*Pg.16 Line 1: the values of albedo given for min and max don't match those in Table 4. I'm confused. Also the precision for RMSE values is probably too great.*

The lines the reviewer is referring states the old and the new threshold values. The new numbers agree with the numbers in table 4. The two albedo values ($\alpha$_min = 0.77 and $\alpha$_max = 0.84) from Dorn et al. (2009) are not listed in table 4. We added the reference to table 2 to omit misunderstanding.

"The adjusted albedo parameters clearly describe the two cloud conditions with higher minimum and maximum values (0.80, 0.88) for overcast conditions and lower values (0.66, 0.79) for clear sky and broken cloud situations compared to the suggested numbers given in the original sea ice albedo scheme from Dorn et al. (2009) with $\alpha$_min = 0.77 and $\alpha$_max = 0.84 (Table 1)."

However, we reduced the number of digits for the RMSE in the text and in table 4:

"The greatest improvement was found for the parameterization of clear sky surface albedo, where the RMSE values for all cases with c_s> 99% reduced from 0.13 to 0.04, and for all data matching c_i> 90% from 0.14 to 0.04. For overcast situations, the RMSE reduces only slightly from 0.06 to 0.05 for c_s> 99% and for cases with c_i> 90%."

**Table 4.** Variation range of minimum and maximum albedo values for snow covered ice and threshold temperature for the adjustment of the sea ice albedo parameterization following Eq. (3) and (4). The albedo and temperature are modified in steps of 0.01 and 0.1°C, respectively. Final fitting values of $\alpha_{min}$, $\alpha_{max}$, and $T_d$ are given for clear/broken cloud and overcast conditions together with the new and old (in brackets) RMSE values.

| | Clear/broken cloud | Overcast |
|---|---|---|
| Range $\alpha_{min}$ | 0.50 to 1.00 | 0.50 to 1.00 |
| Range $\alpha_{max}$ | 0.50 to 1.00 | 0.50 to 1.00 |
| Range $T_d$ (°C) | -5.0 to -0.1 | -5.0 to -0.1 |
| New $\alpha_{min}$ | 0.66 | 0.80 |
| New $\alpha_{max}$ | 0.79 | 0.88 |
| New $T_d$ (°C) | -2.5 | -3.0 |
| RMSE ($c_s > 99\%$) | 0.04 (0.13) | 0.05 (0.06) |
| RMSE ($c_i > 90\%$) | 0.04 (0.14) | 0.05 (0.06) |

---

## Author Comment (AC2) · 26 Apr 2019

**Reply to Reviewer #2:**

**We thank the reviewer for the time and efforts she/he spent reading our manuscript and providing valuable suggestions and advices. Please find below a discussion of the reviewer's comments (italic). Changes/additions made to the text are underlined and given in quotes.**

*Major concerns:*

*The overwhelmingly dominant surface type for this study is snow-covered ice. Should that be reflected in the title?*
We agree with the reviewer, that the validation is mainly attributed to the parameterization performance of snow covered ice. For the season May/June bare ice and melt ponds surface types are still in minority. However, the range of surface albedo values of snow covered ice underlies strong variations due to changing snow reflection properties and a decreasing snow depth which has to be well characterized for a sufficient parameterization. This magnitude of variation is in the same order than for melt ponds. With the planned observations during the MOSAiC campaign in 2020 we will also cover the melt pond season for an extended validation.
Instead of changing the title (which would make it even longer), we pointed out in the abstract that the validation has some restrictions concerning the observed surface types:
"The selected low-altitude (less than 100 m) flight sections of overall 12 flights were performed over surfaces dominated by snow covered ice. It was found that the range of parameterized SIS albedo for individual days is smaller than that of the measurements."
In addition, we added in the Summary and conclusion:
"The presented results are valid for nearly 100% snow covered sea ice."

*The conclusions of this manuscript, as stated p. 18, line 3 provide only limited scientific insight. The details of the model parameterization and the data set being used to validate it are nicely described, but there is not a lot of fresh scientific insight that results.*
The presented "offline" method to evaluate the SIS albedo parameterization in terms of temperature, snow and cloud cover based on airborne measurements is a reasonable and well suited method. It bridges the local observations of ground-based validation data (which only partly represents the variability of surface characteristics)  and satellite comparisons (albedo product derived from multi-day observation and only under cloudless conditions).
By using concurrent measurements as input parameters, the "offline" method allows a validation which is not affected by the uncertainty of modeled parameters (e.g., surface temperature) caused by the complexity in a coupled climate model. From previous studies (e.g. Dorn et al., 2009) it is known that an improved simulation of feedback processes can finally only be obtained by a harmonized combination of improved parameterizations. In a later study we will implement the adapted parameterization into the model, will perform ensemble runs and evaluate statistically the model skills. Also here, the one-year observations during MOSAiC will serve as perfect test bed.
This validation clearly reveals limitations of the current version of the SIS albedo parameterization in HIRHAM-NAOSIM, which as mentioned are the choice of temperature thresholds when reflection properties change significantly, and the illumination dependence. Since a number of other climate models include similar parameterizations, this study may encourage also other modelers to revise their approaches. From the measurement point, it was already well known that clouds have an impact of the magnitude of the surface albedo, but our study reveals directly the effect of including this information on the performance of the SIS albedo parameterization, which is worthwhile to point it out in a publication. We tried to improve the Summary and conclusion section.

*Minor points:*

*Abstract line 5: "The SIS albedo parameterization was tested using measured quantities of the prognostic variables surface temperature and snow depth to calculate the surface albedo and the individual fractions of the ice surface subtypes (snow covered ice, bare ice, and melt ponds) derived from digital camera images taken onboard of the Polar 5/6 aircraft." It would be helpful to include the albedo measurement in this list (broadband? Spectral?).*

We included the observations a little bit earlier here:

"Therefore, the sea ice surface (SIS) albedo parameterization of the coupled regional climate model HIRHAM--NAOSIM was examined against broadband surface albedo measurements performed during the joint ACLOUD (Arctic CLoud Observations Using airborne measurements during polar Day) and PASCAL (Physical feedbacks of Arctic boundary layer, Sea ice, Cloud and AerosoL) campaigns which were performed in May/June 2017 north of Svalbard."

*abstract line 10: "...a temporal bias was observed..." Is this necessarily a temporal bias? It's probably more likely a surface type bias. I doubt the bias depends explicitly on time, but it more likely depends on surface type.*

We exchanged the phrase "temporal" by "time-variable" to emphasize that the bias was variable during the course of the campaign. Nevertheless, the change of surface properties (and consequently the surface albedo) is not instantaneously changing with the increase of temperature. Therefore, one could call it a time-dependent bias.

"Furthermore, a time-variable bias was observed with higher values compared to the modeled SIS albedo (0.88 compared to 0.84 for 29 May 2017) in the beginning of the campaign, and an opposite trend towards the end of the campaign (0.67 versus 0.83 for 25 June 2017)."

*p. 2, line 2: "...the second main contributor." compared to what process?*

We added the lapse rate feedback:

"Pithan and Mauritsen (2014) quantified the strength of various feedback mechanisms contribution to Arctic amplification using climate simulations from the Coupled Model Intercomparison Project Phase 5 (CMIP5; Taylor et al., 2012) and found that the snow/ice albedo feedback is the second main contributor besides the lapse rate feedback."

*p. 2, line 6: "the spread of climate model results with respect to the snow/ice albedo feedback has been discussed" can this be made more specific? I think I understand this sentence is trying to convey that the sensitivity of climate model results to parameters directly related to snow/ice albedo feedback are discussed, but this is not clear.*

The sentence introduces the more specific subsequent paragraph. However, we changed the wording:

"In particular, the spread of climate model results quantifying the snow/ice albedo feedback has been discussed (Qu and Hall, 2014; Thackeray and Fletcher, 2016; Thackeray et al.,2018)."

*Table 1: Where do the min and max values come from? 0.51 - 0.57 seems like a range that I would expect to be biased low.*

The numbers given in Table 1 are suggested by Køltzow (2007) who introduced the sea ice albedo scheme of HIRHAM. The range of the bare ice albedo (0.51 – 0.57) is taken from Table 7:

**Table 7.** Albedo Values for Different Surface Types in the Proposed New Sea Ice Albedo Scheme

| | |
|---|---|
| $\alpha_{DRYSNOW} = 0.84$ | Grenfell and Perovich [1984], Grenfell et al. [1994], Curry et al. [1996] and [Curry et al., 2001] |
| $\alpha_{MELTING\_SNOW} = 0.77$ | Curry et al., 2001, Lindsay and Rothrock [1994] and Perovich et al. [2002a] |
| $\alpha_{BARE\_ICE} = 0.57$ | [Persson et al., 2002; Eicken et al., 1994] |
| $\alpha_{MELTING\_SEA\_ICE} = 0.51$ | Curry et al. [2001] |
| $\alpha_{MELTPONDS}(T_S) = 0.36-0.1$ $(2 + T_S)\, T_S \geq -2°C$ | Tschudi et al. [2001], Perovich and Grenfell [1981], Langleben [1969] and Perovich et al. [2002a] |

(from Køltzow, 2007)

*p. 10, line 14: "This implies that the reflected radiation from side directions has a minor contribution than radiation coming from directly below the aircraft." Please rewrite for clarity– "...has a minor contribution relative to radiation..."?*
We changed the wording:
"This implies that the reflected radiation from side directions has a minor contribution relative to the radiation coming from nadir direction."

*p.12 line 6: delete "it", also please explain what is meant by "structured snow covered ice".*
We adjusted the sentence and used now the term "surface roughness" instead of the phrase "structured snow covered ice". "Surface roughness" is probably more appropriate in this scientific field:
"This results in an increase of surface roughness on 31 May which is also apparent from the digital camera images."

*p. 14, line 2 -3: If I understand correctly, this albedo parameterization does not account for varying grain size and snow depth? That seems like it is important to mention.*
As suggested, we explicitly mentioned it now in a separate sentence:
"In fact, the decrease in SIS albedo is mainly caused by a temporal change (Figure 8b) of the surface properties, as grain size and snow thickness. As obvious from Eq. (3), both parameters are not considered in the SIS albedo parameterization of HIRHAM-NAOSIM."

*p. 14, line 7: "...also the illumination conditions might have an impact on the variation of the surface albedo. Lower SIS albedo values were measured for all cases under cloudless and broken cloud conditions compared to overcast situations with similar surface temperatures ranges." That is expected and it would be helpful to acknowledge that here.*
We gave some references of publication where the illumination dependence was discussed:
"Besides snow property changes, also the illumination conditions might have an impact on the variation of the surface albedo (Choudhury and Chang, 1981; Pirazzini et al., 2015)."

---

## Referee Report (RR1)

**Jakel et al.: Validation of the sea ice surface albedo scheme of the regional climate model HIRHAM-NAOSIM using aircraft measurements during the ACLOUD/PASCAL campaigns**

The authors have done a lot of work and primarily addressed my major concerns from my previous review. Well done! The changes to the figures have made them more understandable too. However, I do have a few minor remaining comments to make assumptions and other conclusions more explicit in the text.

- Cloud focus: The addition of explanation of the cloud impact on broadband albedo is helpful, but there is still some confusion here. I agree, simulating both clouds and radiation in the Arctic is problematic, but I was not suggesting you try to focus those aspects and therefore the addition in section 4 wasn't necessary. What is the "cloudy/overcast" vs. "broken/clear" threshold the sea ice model uses and how is this information conveyed?
- Figure 4: The caption should explicitly state the albedos consider **both** open water and ice. Please change the label "Sea Surface Temperature" to "Surface Skin Temperature." Then in the caption I'd recommend: "(a) Mean skin temperature along flight tracks for selected days … (b) Mean albedo along flight tracks. The surface below the flight track was primarily sea ice covered by snow, but the measurements also consider locations with open water."
- Figure 8: I think there needs to be text to clarify about the magnitude. I'd suggest adding to the sentence about snow type variation something along the lines of: "Furthermore our results indicate that the snow type variation (e.g. change of roughness, grain size, density) that is evident based on the temperature range in Fig.4a or the temporal evolution in Fig.4b is of the same importance for albedo as the illumination effect."
- 22% melt pond fraction: I think you should add text about how this is based on SHEBA and may not be true in the "New" Arctic. I have definitely seen a lot of imagery with pond fraction > 22%.
- Sample size of 50: your explanation makes sense, but I think you should include it in the text as well.

---

## Author Response (AR2)

Reply to the Reviewer:

We thank the reviewer for her/his positive reply.

Please find below a discussion of the reviewer's comments (italic). Changes/additions made to the text are underlined and given in quotes.

*The authors have done a lot of work and primarily addressed my major concerns from my previous review. Well done! The changes to the figures have made them more understandable too. However, I do have a few minor remaining comments to make assumptions and other conclusions more explicit in the text.*

*• Cloud focus: The addition of explanation of the cloud impact on broadband albedo is helpful, but there is still some confusion here. I agree, simulating both clouds and radiation in the Arctic is problematic, but I was not suggesting you try to focus those aspects and therefore the addition in section 4 wasn't necessary. What is the "cloudy/overcast" vs. "broken/clear" threshold the sea ice model uses and how is this information conveyed?*

So far, HIRHAM-NAOSIM is currently tested with a threshold of 0.95. But these tests are still ongoing and are not part of this paper here, since the albedo scheme is run offline in this work. Within the coupled model only the atmospheric part HIRHAM uses the surface albedo to calculate the radiative fluxes, therefore an exchange ("information convey") of such a threshold to the NAOSIM module is not applied.

*• Figure 4: The caption should explicitly state the albedos consider both open water and ice. Please change the label "Sea Surface Temperature" to "Surface Skin Temperature." Then in the caption I'd recommend: "(a) Mean skin temperature along flight tracks for selected days … (b) Mean albedo along flight tracks. The surface below the flight track was primarily sea ice covered by snow, but the measurements also consider locations with open water."*

We adapted Fig. 4 as suggested and changed some text:

[Figure]

**Figure 4.** (a) Mean surface  skin temperature along flight tracks for selected days derived from KT19 measurements onboard of Polar 5 and Polar 6. (b)  Corresponding mean albedo . The error bars give the standard deviation. The surface below the flight tracks was primarily snow covered sea ice, but the measurements also consider locations with open water.

"The temporal development of the standard deviation in Figure 4 shows that the variability of surface skin temperature caused by the contrast between warmer open water and colder sea ice along the individual flight tracks decreases with time."

• *Figure 8: I think there needs to be text to clarify about the magnitude. I'd suggest adding to the sentence about snow type variation something along the lines of:*
*"Furthermore our results indicate that the snow type variation (e.g. change of roughness, grain size, density) that is evident based on the temperature range in Fig.4a or the temporal evolution in Fig.4b is of the same importance for albedo as the illumination effect."*

We adapted the suggestion made by the reviewer in order to better fit into the context.

"Lower SIS albedo values were measured for all cases under cloudless and broken cloud conditions compared to overcast situations with a similar surface temperature. From Figure 8 it is evident that the illumination effect is of the same importance as the snow type variation (e.g. change of roughness, grain size, density) within the observed temperature range."

• *22% melt pond fraction: I think you should add text about how this is based on SHEBA and may not be true in the "New" Arctic. I have definitely seen a lot of imagery with pond fraction > 22%.*

From Fig. 5 of Perovich et al. 2002 we see no larger melt pond fractions than given in surface albedo scheme of HIRHAM-NAOSIM. As already mentioned in the previous reply we are aware of the fact that other measurements show much higher values. The modellers are willing to change this threshold based on new measurements (e.g., from the upcoming MOSAiC campaign). For ACLOUD this threshold will not affect the results, since the observed melt pond fraction is much lower than 22 %.

[Figure]

**Figure 5.** Average values of relative areas of ice, ponds, and leads from May 1998 through October 1998. Over 100 photos were analyzed for each flight.

*From: Perovich, D. K., W. B. Tucker III, and K. A. Ligett, Aerial observations of the evolution of ice surface conditions during summer, J. Geophys. Res. ,107(C10), 8048, doi:10.1029/2000JC000449, 2002.*

Yes, We added some discussion:

"The limitation of the melt pond fraction prevents a complete conversion from snow to melt ponds when temperature is reaching 0°C. The given value of 0.22 refers to SHEBA data derived from aerial imagery (Perovich et al., 2002). More recent data (see e.g., Istomina et al., 2015) indicate that this value is too low, in particular for first year ice that has become more important over the last years. Readjustment of $c_{m,max}$, for instance as a function of the sea-ice age, is envisaged for follow-up studies.. However, for the time frame of the ACLOUD campaign, the threshold of $c_{m,max}$ = 0.22 is appropriate."

• *Sample size of 50: your explanation makes sense, but I think you should include it in the text as well.*

As suggested we added the following:

[revised manuscript text omitted]